# Study on the Calculation Method of Stress in Strong Constraint Zones of the Concrete Structure on the Pile Foundation Based on Eshelby Equivalent Inclusion Theory

**DOI:** 10.3390/ma13173815

**Published:** 2020-08-29

**Authors:** Min Yuan, Dan Zhou, Jian Chen, Xia Hua, Sheng Qiang

**Affiliations:** 1College of Water Conservancy and Hydropower Engineering, Hohai University, Nanjing 210098, China; miny2018@163.com; 2Nanjing R&D Tech Group Co., Ltd., Nanjing 210029, China; czzd@163.com; 3Huai’an Investigation and Design Institute of Water Conservancy, Huai’an 223005, China; hsy.chenjian@163.com; 4College of Engineering and Computer Sciences, Marshall University, Huntington, WV 25755, USA; huax@marshall.edu

**Keywords:** Eshelby, equivalent inclusion, anisotropy, equivalent pile foundation, concrete stress

## Abstract

In view of the strong constraint zones of the concrete structure on the pile foundation, there are some differences between the calculation results of the isotropic equivalent pile foundation by the volume replacement ratio method and the actual engineering. In this paper, referring to the relevant algorithm of rock mass with anchor, the anchor and rock mass are, respectively, compared to pile and surrounding soil foundation. Eshelby equivalent inclusion theory is introduced into the equivalent mechanical model of soil foundation with pile, and a new equivalent pile foundation algorithm considering anisotropic elastic constant is compiled by Fortran. Three kinds of calculation methods are used to calculate the stress field of the concrete structure of the large pump station on the pile foundation during the construction period, and the stress in the strong constraint zones of the concrete structure are mainly analyzed. It is found that the calculation accuracy of Algorithm 3 is the highest, and the calculation results of Algorithm 2 can be modified by the coefficients to achieve the calculation accuracy of Algorithm 3 and the calculation efficiency is actually improved. Finally, the accuracy of the proposed method is verified by the engineering measured data.

## 1. Introduction

At present, many scholars at home and abroad have studied the relationship between composite foundation and parameters such as modulus of elasticity [1,2] and Poisson’s ratio [3,4], considered the influence of volume replacement ratio, foundation stratification, pile diameter, pile length, cushion thickness and other factors on composite foundation [5,6], and also established calculation methods related to composite foundation settlement, pile–soil stress ratio and other aspects [7,8]. However, there are few research studies on the influence of composite foundation on the upper concrete structure, especially the strong constraint zones of the concrete structure. In the process of stress field simulation calculation of large-scale pump stations during construction, there are two main algorithms to deal with the complex foundation with piles. One is Algorithm 1, which is to separate the pile and soil foundation into corresponding finite elements. This algorithm has relatively high calculation accuracy, but the pretreatment process is complex, and the calculation time is long. The other is Algorithm 2, which is the isotropic equivalent pile based on the volume replacement ratio method. This algorithm simplifies the pretreatment process of complex foundation and reduces the calculation time, so it is often used. However, the calculation accuracy of the algorithm is not ideal, and there are some differences between the algorithm and the engineering practice. The practical engineering shows that, especially for the strong constraint zones [9,10] of the concrete structure on the pile foundation, some cracks still appear after taking corresponding crack prevention measures. The main reason is that the calculation results of Algorithm 2 cannot accurately predict the stress value of the concrete structure, which leads to unreasonable crack prevention measures. Therefore, whether it is reasonable and effective to calculate the stress in strong constraint zones of the concrete structure on the pile foundation by Algorithm 2 needs further exploration.

In this paper, referring to the relevant algorithm of anchored rock mass [11] and aiming at the problems of the above composite foundation, the pile and surrounding soil foundation are compared to the anchor and rock mass, respectively, and the Eshelby equivalent inclusion theory is introduced into the equivalent mechanical model of soil foundation with pile, that is the new algorithm of the anisotropic pile foundation (Algorithm 3). The stress field of a large pump station with complicated pile foundation during the construction period is simulated by using a three-dimensional finite element simulation program compiled by Fortran [12,13,14], and the stress in the strong constraint zones of the concrete structure is emphatically analyzed. Through the comprehensive comparison of the calculation results of the three algorithms, it can be seen that the calculation accuracy of Eshelby equivalent inclusion theory (Algorithm 3) is significantly higher than that of the isotropic equivalent pile foundation (Algorithm 2) with the volume replacement ratio method. At the same time, it is found that there is a certain ratio rule between the maximum values of the first principal stress obtained by the two algorithms, and the corresponding correction coefficient of stress calculation value is obtained according to the rule. The correction coefficient of the stress calculation value keeps the advantages of simple pretreatment process and high efficiency of Algorithm 2 and, furthermore, improves the calculation accuracy relatively. Through the numerical calculation and the engineering measurement data, the rationality of the calculation method and the correctness of the correction coefficient introduced in this paper are verified, which can provide some references for similar projects.

## 2. Eshelby Equivalent Inclusion Theory 

Eshelby equivalent inclusion theory is widely used in mixed materials [14,15,16]. In addition, Xu et al. [17] and Hu et al. [18] studied the elastic constants based on the Eshelby equivalent inclusion theory, while Liu et al. [19] studied the thermal conductivity based on the Eshelby equivalent inclusion theory. According to Eshelby’s derivation, the strain produced by uniform eigenstrain in ellipsoid is also uniform. That is to say, when an isotropic matrix containing an ellipsoidal inclusion is subjected to an external force, if there is a uniform strain in the matrix, the strain in the ellipsoidal inclusion is also uniform, which can be expressed as [11]:(1)$$\varepsilon_{in}=S_{inkl}\cdot \varepsilon_{kl}^{*}$$
where $S_{inkl}$ is the Eshelby fourth order tensor, the Eshelby tensor is a constant tensor only related to the Poisson’s ratio of the matrix and the shape of the inclusions, with symmetry for the coordinates *i*, *n*, *k* and *l*, that is:(2)$$S_{inkl}=S_{nikl}=S_{inlk}$$

The symbols a_1_, a_2_, a_3_ are introduced. a_1_, a_2_, a_3_ are the three principal semi axes of the ellipsoidal inclusion. When the ellipsoid inclusion is an approximate cylinder, that is a_2_ = a_3_, a_1_ → ∞, the Eshelby tensor can be expressed as [11]:(3)$$\begin{array}{l} S_{1111}=0\\ S_{2222}=S_{3333}=\frac{5-4v_{0}}{8\left(1-v_{0}\right)}\\ S_{2233}=S_{3322}=\frac{4v_{0}-1}{8\left(1-v_{0}\right)}\\ S_{1122}=S_{1133}=0\\ S_{2211}=S_{3311}=\frac{v_{0}}{2\left(1-v_{0}\right)}\\ S_{1212}=S_{1313}=\frac{1}{4}\\ S_{2323}=\frac{3-4v_{0}}{8\left(1-v_{0}\right)} \end{array}\}$$

## 3. Establishment of Anisotropic Equivalent Mechanical Model of Soil Foundation with Piles

### 3.1. Elastic Stress–Strain Relationship

Based on the hypothesis and analysis of the Mori–Tanaka method, the mechanical properties of the soil foundation with the piles are compared to orthotropic anisotropy. Anisotropy is one of the most important characteristics of engineering geotechnical materials. Zhang et al. [20] and Li et al. [21] studied the relationship between the dynamic response of slab and foundation; Yang et al. [22] and Reccia et al. [23] calculated the bearing capacity and settlement of anisotropic foundation. In the elastic stage, the stress–strain relationship can be expressed as:(4){ε}=[S]{σ} or {σ}=[C]{ε}
where $\left\{\varepsilon \right\}={\left\{\varepsilon_{11},\varepsilon_{22},\varepsilon_{33},\varepsilon_{23},\varepsilon_{31},\varepsilon_{12}\right\}}^{T}$, $\left\{\sigma \right\}={\left\{\sigma_{11},\sigma_{22},\sigma_{33},\sigma_{23},\sigma_{31},\sigma_{12}\right\}}^{T}$. [S] is the flexibility matrix, [C] is the stiffness matrix, $\left[S\right]={\left[C\right]}^{-1}$ is the inverse matrix of the stiffness matrix [C]. For orthotropic anisotropic materials, the flexibility matrix is:(5)$$\left[S\right]=\left\{\begin{array}{cccccc} S_{11} & S_{12} & S_{13} & 0 & 0 & 0\\ S_{21} & S_{22} & S_{23} & 0 & 0 & 0\\ S_{31} & S_{32} & S_{33} & 0 & 0 & 0\\ 0 & 0 & 0 & S_{44} & 0 & 0\\ 0 & 0 & 0 & 0 & S_{55} & 0\\ 0 & 0 & 0 & 0 & 0 & S_{66} \end{array}\right\}$$

There are nine independent flexibility coefficients in the above equations. In engineering, the engineering elastic constant is often used to express the elastic constant of composite materials. The relationship between the engineering elastic constant and the flexibility coefficient can be expressed as [24]:(6)$$\left[S\right]=\left[\begin{array}{cccccc} \frac{1}{E_{11}} & \frac{-\nu_{12}}{E_{22}} & \frac{-\nu_{13}}{E_{33}} & 0 & 0 & 0\\ \frac{-\nu_{21}}{E_{11}} & \frac{1}{E_{22}} & \frac{-\nu_{23}}{E_{33}} & 0 & 0 & 0\\ \frac{-\nu_{31}}{E_{11}} & \frac{-\nu_{32}}{E_{22}} & \frac{1}{E_{33}} & 0 & 0 & 0\\ 0 & 0 & 0 & \frac{1}{G_{23}} & 0 & 0\\ 0 & 0 & 0 & 0 & \frac{1}{G_{31}} & 0\\ 0 & 0 & 0 & 0 & 0 & \frac{1}{G_{12}} \end{array}\right]$$
where $E_{11},E_{22},E_{33},v_{12},v_{13},v_{21},v_{23},v_{31},v_{32},G_{23},G_{31},G_{12}$ make up the mechanical elastic constant of soil foundation with piles. For orthotropic materials, there are only nine independent elastic parameters, because $S_{ij}=S_{ji}$, there are: (7)$$\begin{array}{l} \frac{v_{21}}{E_{11}}=\frac{v_{12}}{E_{22}}\\ \frac{v_{31}}{E_{11}}=\frac{v_{13}}{E_{33}}\\ \frac{v_{32}}{E_{22}}=\frac{v_{23}}{E_{33}} \end{array}\},\;\mathrm{that}\;\mathrm{is}\;\frac{v_{ij}}{E_{j}}=\frac{v_{ji}}{E_{i}},\;(i,j\;=\;1,\;2,\;3,\;\mathrm{but}\;i\;\neq \;j)$$
where $v_{ij}$ are six in total, but three of them can be represented by another three Poisson’s and $E_{11},E_{22},E_{33}$. Generally, the three relations in Equation (7) are called Maxwell’s theorem.

According to the matrix analysis, {ε}=[S]{σ} can be expressed by the following two expressions:(8)$$\left\{\begin{array}{c} \varepsilon_{11}\\ \varepsilon_{22}\\ \varepsilon_{33} \end{array}\right\}=\left[-\begin{array}{ccc} \frac{1}{E_{11}} & -\frac{\nu_{12}}{E_{22}} & -\frac{\nu_{13}}{E_{33}}\\ \frac{-\nu_{21}}{E_{11}} & \frac{1}{E_{22}} & -\frac{\nu_{23}}{E_{33}}\\ -\frac{\nu_{31}}{E_{11}} & -\frac{\nu_{32}}{E_{22}} & \frac{1}{E_{33}} \end{array}\right]\left\{\begin{array}{c} \sigma_{11}\\ \sigma_{22}\\ \sigma_{33} \end{array}\right\}$$
(9)$$\left\{\begin{array}{c} \varepsilon_{23}\\ \varepsilon_{31}\\ \varepsilon_{12} \end{array}\right\}=\left[\begin{array}{ccc} \frac{1}{G_{23}} & 0 & 0\\ 0 & \frac{1}{G_{31}} & 0\\ 0 & 0 & \frac{1}{G_{12}} \end{array}\right]\left\{\begin{array}{c} \sigma_{23}\\ \sigma_{31}\\ \sigma_{12} \end{array}\right\}$$

### 3.2. Determination of Elastic Constants

Zhao introduced the theory of Eshelby’s equivalent inclusion into the determination of elastic constant of rock mass with anchor [11]; this paper refers to its practice and introduces the theory into the determination of elastic constant of soil foundation with pile.

According to the analysis of elastic mechanics, in the composite material with the pile and soil foundation, the elastic constants of the soil foundation and pile can be expressed as [11]:(10)$$H_{inkl}^{0}=\lambda_{0}\cdot \delta_{in}\cdot \delta_{kl}+\mu_{0}\left(\delta_{in}\cdot \delta_{kl}+\delta_{il}\cdot \delta_{nk}\right)$$
(11)$$H_{inkl}^{1}=\lambda_{1}\cdot \delta_{ij}\cdot \delta_{kl}+\mu_{1}\left(\delta_{ij}\cdot \delta_{kl}+\mu_{1}\cdot \delta_{il}\cdot \delta_{jk}\right)$$
where $H_{inkl}^{0}$ and $H_{inkl}^{1}$ are the tensors of elastic coefficient of soil foundation and pile, respectively, and $\lambda_{0}$ and $\mu_{0}$ are the lame constants of soil foundation, and $\lambda_{0}=\frac{v_{0}\cdot E_{0}}{\left(1+v_{0}\right)\left(1-2v_{0}\right)}$, $\mu_{0}=\frac{E_{0}}{2\left(1+v_{0}\right)}$, $v_{0}$, $E_{0}$ are the Poisson’s ratio and elastic modulus of soil foundation, respectively. $\delta_{ij}$, $\delta_{kl}$, $\delta_{il}$, $\delta_{jk}$ are tensor symbols. 

A homogeneous stress is applied on the boundary of the composite material with the pile and soil foundation, and an isotropic material with the same shape and size and the same elastic mechanical properties as that of the soil (matrix) material in the pile–soil foundation is set. Under the same external force, the stress–strain relationship of the soil foundation can be expressed as [11]:(12)$$\sigma^{0}=C_{0}\cdot \varepsilon^{0}$$
where $C_{0}$ is the elastic constant of the soil foundation with piles. Due to the existence of piles, the average strain produced in the soil foundation is not equal to $\varepsilon^{0}$, and the interaction between the piles will produce a disturbance strain $\tilde{\varepsilon }$, and the corresponding disturbed stress is $\tilde{\sigma }$. Therefore, the constitutive relation of the soil foundation with piles can be given as [11]:(13)$$\sigma^{\left(0\right)}=\sigma^{0}+\tilde{\sigma }=C_{0}\varepsilon^{0}+C_{0}\tilde{\varepsilon }=C_{0}\left(\varepsilon^{0}+\tilde{\varepsilon }\right)$$

Eshelby equivalent inclusion theory points out that the disturbed strain field caused by different mechanical properties of materials can be simulated by the disturbed field generated by the intrinsic strain $\varepsilon^{*}$ in the inclusion domain. That is to say, the inclusion and the matrix can be regarded as the same material, which can be expressed as [11]:(14)$$\sigma^{\left(1\right)}=\sigma^{0}+\tilde{\sigma }+\sigma^{\prime }=C_{1}\left(\varepsilon^{0}+\tilde{\varepsilon }+\varepsilon^{\prime }\right)=C_{0}\left(\varepsilon^{0}+\tilde{\varepsilon }+\varepsilon^{\prime }-\varepsilon^{*}\right)$$
where $C_{1}$ is the elastic constant tensor of the pile in the soil foundation with pile, and $\varepsilon^{*}$ is the equivalent intrinsic strain of the pile in the soil foundation with pile. $\varepsilon^{\prime }$ and $\sigma^{\prime }$ are the disturbance stress and strain, respectively, due to the existence of piles. Follow the Eshelby equivalent inclusion theory to export the results:(15)$$\varepsilon^{\prime }=S\cdot \varepsilon^{*}$$
where S is the Eshelby fourth order tensor.

Substituting Equations (13) and (14) into (15) can be expressed as:(16)$$\sigma^{\prime }=C_{0}\left(\varepsilon^{\prime }-\varepsilon^{*}\right)=C_{0}\left(S-I\right)\varepsilon^{*}$$
where *I* is the fourth order tensor.

We assume that n is the volume replacement ratio, that is, the volume ratio of pile to soil foundation with pile. According to the principle of equivalent inclusion, the average stress σ of the composite material with pile and soil foundation is equal to the homogeneous stress $\sigma^{0}$ applied on the boundary.
(17)$$\sigma =\sigma^{0}=\left(1-n\right)\cdot \sigma^{0}+n\cdot \sigma^{\left(1\right)}$$

Equations (16) and (17) can be expressed as:(18)$$\begin{array}{l} \tilde{\sigma }=-n\cdot \sigma^{\prime }\\ \tilde{\varepsilon }=-n_{1}\cdot \left(\varepsilon^{\prime }-\varepsilon^{*}\right)=-n\cdot \left(S-I\right)\varepsilon^{*} \end{array}\}$$

Substituting Equations (16) and (18) into (14), respectively, can be expressed as:(19)$$\varepsilon^{*}={\left\{C_{0}+\left(C_{1}-C_{0}\right)\cdot \left[nI+\left(1-n\right)S\right]\right\}}^{-1}\cdot \left(C_{0}-C_{1}\right)\cdot \varepsilon^{0}$$

Suppose $P={\left\{C_{0}+\left(C_{1}-C_{0}\right)\cdot \left[nI+\left(1-n\right)S\right]\right\}}^{-1}\cdot \left(C_{0}-C_{1}\right)$, the above equations can be expressed as:(20)$$\varepsilon^{*}=P\cdot \varepsilon^{0}$$

For soil foundation with piles, substituting Equations (10), (11), (15) and (18) into Equation (14) can be expressed as [11]:(21)$$\left[\begin{array}{ccc} D_{1} & D_{2} & D_{3}\\ D_{4} & D_{5} & D_{6}\\ D_{7} & D_{8} & D_{9} \end{array}\right]\left\{\begin{array}{c} \varepsilon_{11}^{*}\\ \varepsilon_{22}^{*}\\ \varepsilon_{33}^{*} \end{array}\right\}+\left[\begin{array}{ccc} L_{1} & 1 & 1\\ 1 & L_{1} & 1\\ 1 & 1 & L_{1} \end{array}\right]\left\{\begin{array}{c} \varepsilon_{11}^{0}\\ \varepsilon_{22}^{0}\\ \varepsilon_{33}^{0} \end{array}\right\}=0$$
where $\left[\begin{array}{ccc} D_{1} & D_{2} & D_{3}\\ D_{4} & D_{5} & D_{6}\\ D_{7} & D_{8} & D_{9} \end{array}\right]$ is the elastic modulus matrix of the soil foundation with pile, $\left[\begin{array}{ccc} L_{1} & 1 & 1\\ 1 & L_{1} & 1\\ 1 & 1 & L_{1} \end{array}\right]$ is the elastic modulus matrix of the homogeneous soil foundation, $\left\{\begin{array}{c} \varepsilon_{11}^{*}\\ \varepsilon_{22}^{*}\\ \varepsilon_{33}^{*} \end{array}\right\}$ is the equivalent intrinsic strain of the soil foundation with pile, and $\left\{\begin{array}{c} \varepsilon_{11}^{0}\\ \varepsilon_{22}^{0}\\ \varepsilon_{33}^{0} \end{array}\right\}$ is the homogeneous soil foundation strain, where: $$\begin{array}{c} D_{1}=n\cdot L_{1}+L_{2}+\left(1-n\right)\cdot \left(S_{2211+}S_{3311}\right)\\ D_{2}=n+L_{3}+\left(1-n\right)\cdot \left(S_{2222+}S_{3322}\right)\\ D_{3}=n+L_{3}+\left(1-n\right)\cdot \left(S_{2233+}S_{3333}\right)\\ D_{4}=n+L_{3}+\left(1-n\right)\cdot \left(L_{1}\cdot S_{2211+}S_{3311}\right)\\ D_{5}=n\cdot L_{1}+L_{2}+\left(1-n\right)\cdot \left(L_{1}\cdot S_{2222+}S_{3322}\right)\\ D_{6}=n+L_{3}+\left(1-n\right)\cdot \left(L_{1}\cdot S_{2233+}S_{2211}\right)\\ D_{7}=n+L_{3}+\left(1-n\right)\cdot \left(L_{1}\cdot S_{3322+}S_{2222}\right)\\ D_{8}=n+L_{3}+\left(1-n\right)\cdot \left(L_{1}\cdot S_{3322+}S_{2222}\right)\\ D_{9}=n\cdot L_{1}+L_{2}+\left(1-n\right)\cdot \left(L_{1}\cdot S_{3333+}S_{2233}\right) \end{array}$$
where $L_{1}=1+\frac{2\left(\mu_{1}-\mu_{0}\right)}{\left(\lambda_{1}-\lambda_{0}\right)}$, $L_{2}=\frac{\left(\lambda_{1}+2\mu_{0}\right)}{\left(\lambda_{1}-\lambda_{0}\right)}$, $L_{3}=\frac{\lambda_{0}}{\left(\lambda_{1}-\lambda_{0}\right)}$.

Equation (20) shows the relationship between the equivalent effect $\xi_{ij}^{*}$ of the composite material with pile and soil foundation and soil foundation $\xi_{ij}^{0}$, according to $\varepsilon_{ij}^{*}=P\cdot \varepsilon_{ij}^{0}$ can be expressed as [11]:(22)$$\left\{\begin{array}{c} \varepsilon_{11}^{*}\\ \varepsilon_{22}^{*}\\ \varepsilon_{33}^{*} \end{array}\right\}=\frac{1}{p}\left\{\begin{array}{ccc} Q_{1} & Q_{2} & Q_{3}\\ Q_{4} & Q_{5} & Q_{6}\\ Q_{7} & Q_{8} & Q_{9} \end{array}\right\}\left\{\begin{array}{c} \varepsilon_{11}^{0}\\ \varepsilon_{22}^{0}\\ \varepsilon_{33}^{0} \end{array}\right\}$$
where:$$\begin{array}{c} Q_{1}=L_{1}\cdot \left(D_{6}\cdot D_{8}-D_{5}\cdot D_{9}\right)+D_{3}\cdot \left(D_{5}-D_{8}\right)+D_{2}\cdot \left(D_{9}-D_{5}\right)\\ Q_{2}=L_{1}\cdot \left(D_{2}\cdot D_{9}-D_{3}\cdot D_{8}\right)+D_{6}\cdot \left(D_{8}-D_{2}\right)+D_{5}\cdot \left(D_{3}-D_{9}\right)\\ Q_{3}=L_{1}\cdot \left(D_{3}\cdot D_{5}-D_{2}\cdot D_{6}\right)+D_{8}\cdot \left(D_{6}-D_{3}\right)+D_{9}\cdot \left(D_{2}-D_{5}\right)\\ Q_{4}=L_{1}\cdot \left(D_{4}\cdot D_{9}-D_{6}\cdot D_{7}\right)+D_{1}\cdot \left(D_{6}-D_{9}\right)+D_{3}\cdot \left(D_{7}-D_{4}\right)\\ Q_{5}=L_{1}\cdot \left(D_{3}\cdot D_{7}-D_{1}\cdot D_{9}\right)+D_{4}\cdot \left(D_{9}-D_{3}\right)+D_{6}\cdot \left(D_{1}-D_{7}\right)\\ Q_{6}=L_{1}\cdot \left(D_{1}\cdot D_{6}-D_{3}\cdot D_{4}\right)+D_{9}\cdot \left(D_{4}-D_{1}\right)+D_{7}\cdot \left(D_{3}-D_{6}\right)\\ Q_{7}=L_{1}\cdot \left(D_{5}\cdot D_{7}-D_{4}\cdot D_{8}\right)+D_{2}\cdot \left(D_{4}-D_{7}\right)+D_{1}\cdot \left(D_{8}-D_{5}\right)\\ Q_{8}=L_{1}\cdot \left(D_{1}\cdot D_{8}-D_{2}\cdot D_{7}\right)+D_{5}\cdot \left(D_{7}-D_{1}\right)+D_{4}\cdot \left(D_{2}-D_{8}\right)\\ Q_{9}=L_{1}\cdot \left(D_{2}\cdot D_{4}-D_{1}\cdot D_{5}\right)+D_{7}\cdot \left(D_{5}-D_{2}\right)+D_{8}\cdot \left(D_{1}-D_{4}\right)\\ P=D_{1}\cdot \left(D_{5}\cdot D_{9}-D_{6}\cdot D_{8}\right)+D_{2}\cdot \left(D_{6}\cdot D_{7}-D_{4}\cdot D_{9}\right)+D_{3}\cdot \left(D_{4}\cdot D_{8}-D_{5}\cdot D_{7}\right) \end{array}$$

Further derivation can be expessed as [11]: (23)$$\varepsilon_{12}^{*}=\frac{-\left(\mu_{1}-\mu_{0}\right)}{\mu_{0}+\left(\mu_{1}-\mu_{0}\right)\cdot \left[n+2\left(1-n\right)\cdot S_{1212}\right]}\cdot \varepsilon_{12}^{0}$$
(24)$$\varepsilon_{13}^{*}=\frac{-\left(\mu_{1}-\mu_{0}\right)}{\mu_{0}+\left(\mu_{1}-\mu_{0}\right)\cdot \left[n+2\left(1-n\right)\cdot S_{1313}\right]}\cdot \varepsilon_{13}^{0}$$
(25)$$\varepsilon_{12}^{*}=\frac{-\left(\mu_{1}-\mu_{0}\right)}{\mu_{0}+\left(\mu_{1}-\mu_{0}\right)\cdot \left[n+2\left(1-n\right)\cdot S_{2323}\right]}\cdot \varepsilon_{23}^{0}$$

Through the above derivation, the relationship between the equivalent strain of the soil foundation with pile and the intrinsic strain of the soil foundation is obtained. Furthermore, the corresponding engineering elastic constants in the flexibility matrix of soil foundation with pile can be obtained.

(1)Axial elastic modulus of $E_{11}$ is:(26)$$E_{11}=\frac{\varepsilon_{11}^{0}}{\varepsilon_{11}^{0}+n\cdot \varepsilon_{11}^{*}}\cdot E_{0}=\frac{E_{0}}{1+n\cdot \left[Q_{1}-v_{0}\cdot \left(Q_{2}+Q_{3}\right)\right]/P}$$(2)The elastic modulus of $E_{22}$ and $E_{33}$ of the soil foundation with piles are equal along the radius direction of the piles. Radial elastic modulus of $E_{22}$ and $E_{33}$ are:(27)$$E_{22}=E_{33}=\frac{E_{0}}{1+n\cdot \left[Q_{5}-v_{0}\cdot \left(Q_{4}+Q_{6}\right)\right]/P}$$(3)Axial shear modulus of $G_{23}$:(28)$$G_{23}=1+\frac{G_{0}}{2\left(1-n\right)S_{2323}+G_{0}/\left(G_{1}-G_{0}\right)}$$The shear modulus of $G_{12}$ and $G_{13}$ of the soil foundation with piles are equal along the radius direction of the piles. Radial shear modulus of $G_{12}$ and $G_{13}$ are: (29)$$G_{12}=G_{13}=1+\frac{G_{0}}{2\left(1-n\right)S_{1212}+G_{0}/\left(G_{1}-G_{0}\right)}$$(4)The Poisson’s ratios of $v_{12}$ and $v_{13}$ of the soil foundation with piles are equal along the axial direction of the piles. Axial Poisson’s ratios of $v_{12}$ and $v_{13}$ are: (30)$$v_{12}=v_{13}=\frac{v_{0}-n\cdot \left[Q_{4}-v_{0}\cdot \left(Q_{5}+Q_{6}\right)\right]/P}{1+n\cdot \left[Q_{1}-v_{0}\cdot \left(Q_{2}+Q_{3}\right)\right]/P}$$According to the derivation of Maxwell’s theorem in Equation (7), the following can be expressed as:(31)$$v_{21}=v_{12}\left(\frac{E_{22}}{E_{11}}\right)$$
(32)$$v_{31}=v_{13}\left(\frac{E_{33}}{E_{11}}\right)$$(5)Radial Poisson’s ratio of $v_{23}$:(33)$$v_{23}=\frac{v_{0}-n\cdot \left[Q_{6}-v_{0}\cdot \left(Q_{4}+Q_{5}\right)\right]/P}{1+n\cdot \left[Q_{9}-v_{0}\cdot \left(Q_{7}+Q_{8}\right)\right]/P}$$

According to the derivation of Maxwell’s theorem in Equation (7):(34)$$v_{32}=v_{23}\left(\frac{E_{33}}{E_{22}}\right)$$

From $E_{22}$ = $E_{33}$, Equation (35) is expressed as:(35)$$v_{32}=v_{23}$$

From the above, nine mechanical parameters in the flexibility matrix of the soil foundation with piles are derived, and the constitutive relation of the soil foundation with piles in the elastic stage is established. Where $E_{22}$ = $E_{33}$, $v_{12}$ = $v_{13}$, $v_{32}$ = $v_{23}$, $G_{12}$ = $G_{13}$.

## 4. Simulation Calculation Model

### 4.1. Calculation Model

In this paper, a finite element model of concrete slab on soil foundation with piles is established in Figure 1, and we use a 4-node tetrahedral solid element in the calculation model. The three algorithms mentioned above are used to calculate the stress of the concrete slab during the construction period, respectively. In order to avoid the influence of other factors on the calculation results, the three algorithms will adopt a unified calculation grid. However, in Algorithm 2 and Algorithm 3, pile and soil foundation are equivalent to the same material, which is isotropic in Algorithm 2 while anisotropic in Algorithm 3.

Due to the symmetry of the structure, the calculation model takes half of the structure, and the size of the calculation model is as follows:

The foundation: 4.1 × 3.0 × 6.0 = 73.80 m^3^, and the concrete structure on the soil foundation with piles: 4.1 × 3.0 × 1.0 = 12.30 m^3^; Single pile: 0.2 × 0.2 × 6.0 = 0.24 m^3^, quantity: 6.0 × 8.0 = 48.0; Volume replacement ratio: n = 48 × 0.24 ÷ 73.80 = 0.156.

### 4.2. Load Application

The load applied at one time, P = ρgV = 2261 × 9.8 × 12.71 = 2.82 × 10^5^ N, and the approximate value of P in the calculation process is 3.0 × 10^5^ N. It should be noted that this is a layer of 1-meter thick pouring block. Assuming that the superstructure height is 6 meters, the pouring is completed in six times, and the pouring interval is 5 days. A total of six loads were applied to simulate the pouring process of the superstructure. The cumulative total loads on the first day, second day, third day, fourth day, fifth day and sixth day were 3.0 × 10^5^ N, 6.0 × 10^5^ N, 9.0 × 10^5^ N, 1.2 × 10^6^ N, 1.5 × 10^6^ N and 1.8 × 10^6^ N, respectively, in Figure 2.

### 4.3. Feature Points Selection

The research objective is to compare the correctness of the stress in the strong constraint zones of the upper concrete structure of pile foundation under different algorithms. The feature points are taken in the upper concrete structure to observe and compare the first principal stress. A total of eight series of feature points are selected in Figure 3, respectively, corresponding to 1, 2, 3, 4, 5, 6, 7 and 8 in Figure 3. Each series of feature points includes a, b, c, d, e and f from the bottom to the top. For example, 1a, 1b, 1c, 1d, 1e and 1f correspond to the feature points of series 1. In this paper, the elastic modulus of the equivalent pile foundation is greater than the minimum elastic modulus of the class V rock mass of the dam foundation in the design code for the concrete gravity dam (SL319-2005). Referring to this code, the value range of the foundation’s strong constraint zones in the height direction is 0.0–0.2 times of the longest side of the pouring block. In this paper, the length of the long side of the pouring block is 6.0 m, so the strong constraint zones of the foundation are 0.0–1.2 m in the height direction.

### 4.4. Calculation Parameters 

The main mechanical parameters of concrete and soil are shown in Table 1.

### 4.5. Boundary Condition

In the stress field simulation calculation, the four sides and the bottom surface of the foundation are applied to normal constraints, and the upper surface is a free boundary. The symmetry plane of the structure is applied to a normal constraint, and the other surfaces are free boundaries. 

## 5. Calculation Cases

The specific calculation conditions are as follows:

Case 1The pile and soil foundation are simulated as concrete and soil materials, respectively, (Algorithm 1);Case 2Case 2 The isotropic equivalent pile based on the volume replacement ratio method (Algorithm 2);Case 3Case 3 The anisotropic equivalent pile based on the volume replacement ratio method (Algorithm 3).

## 6. Calculation Results and Analysis

According to Table 2’s maximum values of the first principal stress of the series feature points under three algorithms, it can be seen that for most of the feature points, the difference between the calculation results of Algorithm 1 and Algorithm 2 is larger. However, the results of Algorithm 3 are closer to those of Algorithm 1, and the error between them is smaller. 

For the internal points with a distance of 0.5 m or more away from the free surface, they are the three types of feature points a, b and c in series 1, 2, 3 and 7, respectively. For the maximum values of the first principal stress, the calculation result of Algorithm 2 is greater than that of Algorithm 1. For the two types of feature points e and f in this series, the calculation results are opposite. For the other surface points less than 0.5 m away from the free face, the calculation results of Algorithm 2 are basically less than that of Algorithm 1 for the maximum values of the first principal stress.

Therefore, in order to further explore the relationship between the maximum values of the first principal stress at the internal points of 0.5 m or more away from the free surface and the surface points of 0.5 m or less away from the free surface, this paper studies the ratio of the calculation result of the maximum values of the first principal stress between Algorithm 3 and Algorithm 2 (hereinafter referred to as the ratio α). It can be seen from Figure 4 that for the internal points with a distance of 0.5 m or more away from the free surface, for example, the corresponding ratio α of the two types of feature points b and c in series 1, 2, 3 and 7 floats up and down at 0.73, while the corresponding ratio α of the feature point a in series 1, 2, 3 and 7 floats up and down at 0.15. As the position of a characteristic point is too close to the contact surface, this paper does not consider the contact factor’s influence on the maximum values of the first principal stress, the influence will not be analyzed later. It can also be seen from Figure 4 that the difference between the calculation results of Algorithm 2 and that of Algorithm 1 is relatively large for the three types of feature points a, b and c in series 1, 2, 3 and 7 in the vertical direction. For the surface points less than 0.5 m away from the free surface, such as the feature points in series 4, 5, 6 and 8 as shown in Figure 5. No matter they are near or far away from the contact surface, the ratio α floats up and down at 1.33, which shows that the contact surface has little effect on its ratio at this time.

As shown in Figure 6, for the ratio α in the horizontal direction, the ratio α of the two types of feature points e and f in the upper layer floats around 1.33, and for the two types of feature points b and c in the lower layer, the ratio α of the internal points of 0.5 m or more away from the free surface floats around 0.73, while the ratio α jumps to about 1.33 as the distance away from the free surface gets closer.

It is found that this characteristic accords with the general rule, that is, the feature points of the strong constraint zones of the concrete structure on the pile foundation, if the distance away from the free surface is greater than or equal to 0.5 m, the ratio of the maximum values of the first principal stress calculated according to the result of Algorithm 3 to the result of Algorithm 2 is about 0.73. However, if the distance from the temporary surface is less than 0.5 m, the ratio is about 1.33.

In conclusion, the following correction method of stress calculation value is obtained, which can be used to modify the calculation result of stress in the strong constraint zones of concrete structure in Algorithm 2.
(36)$$\sigma_{1\max}=\{\begin{array}{l} \alpha_{1}\cdot {\sigma^{\prime }}_{1\max},\Omega_{1}\\ \alpha_{2}\cdot {\sigma^{\prime }}_{1\max},\Omega_{2} \end{array}$$
where ${\sigma^{\prime }}_{1\max}$ is the maximum value of the first principal stress calculated according to Algorithm 2. 

The zones greater than or equal to 0.5 m away from the free surface are expressed by $\Omega_{1}$, the zones less than 0.5 m away from the free surface are expressed by $\Omega_{2}$. For $\Omega_{1}$ and $\Omega_{2}$, the recommended corresponding correction coefficients $\alpha_{1}$ and $\alpha_{2}$ range from 0.72 to 0.76 and 1.32 to 1.36, respectively. $\sigma_{1\max}$ is expressed as the final value of the maximum value of the first principal stress obtained after the correction method according to the calculated value of the stress. 

To further verify the rationality of the numerical simulation, we apply the theory [11] to practical engineering and compare the theoretical calculation data with the actual data in Section 7. 

## 7. Engineering Verification

### 7.1. Finite Element Model and Feature Point Location

In order to further improve the calculation efficiency and accuracy of three algorithms, we use an 8-node hexahedron solid element in all the calculation models in this chapter. On the other hand, in order to improve the demonstration, the feature points selected in the engineering projects are all located in the strong constraint zones of the concrete structure, the details are as follows:

The whole finite element model of the inlet section of Xiepu pump station is shown in Figure 7a. The total number of units is 138,836 and the total number of nodes is 151,309. The elements of the pile are shown in Figure 7b, the elements of the inlet section are shown in Figure 7c and the location of feature point 1 is shown in Figure 7d.

The whole finite element model of the outlet section of Xiepu pump station is shown in Figure 8a. The total number of units is 143,969 and the total number of nodes is 160,689. The elements of the pile are shown in Figure 8b, the elements of the outlet section are shown in Figure 8c and the location of feature point 2 is shown in Figure 8d.

The whole finite element model of the outlet section of Lianghu pump station is shown in Figure 9a. The total number of units is 86,836 and the total number of nodes is 100,637. The elements of the pile are shown in Figure 9b, the elements of the outlet section are shown in Figure 9c, the location of feature points 3 and 4 are shown in Figure 9d,e, respectively. 

The origin of coordinates is located in the inlet channel, the Z axis is vertical upward, the X axis points to the flow direction, and the Y axis points to the left bank according to the right spiral rule.

### 7.2. Calculation Results and Analysis

It can be seen from Figure 10 and Figure 11 that the calculation results of Algorithm 1 and Algorithm 3 are close to the measured values, while the error of Algorithm 2 relative to the measured values is large, and the maximum relative error is nearly 41%. It can be seen that the results calculated simply according to the volume replacement ratio method are different from the actual results. It can be seen from Table 3 that among the four feature points selected by the Xiepu pump station and the Lianghu pump station. The ratio of the maximum values of the first principal stress of the calculation result of Algorithm 3, to the maximum values of the first principal stress of the calculation result of Algorithm 2, namely, the ratio α, can be observed. For internal points (1 and 3) greater than or equal to 0.5 m away from the free surface, the ratio α is 0.72 and 0.74, respectively—all of which are within the variation range of the correction coefficient $\alpha_{1}$. For the internal points (2 and 4) less than 0.5 m away from the free surface, the ratio α is 1.34 and 1.35, respectively—all of which are within the variation range of the correction coefficient $\alpha_{2}$. 

Compared with Algorithm 2 and Algorithm 1, Algorithm 3 has the advantages of higher precision and simpler pretreatment process, respectively, but its corresponding stress–strain relationship is complex, and the calculation efficiency is relatively reduced. Therefore, on the basis of Algorithm 2, the correction coefficient of stress calculation value (Equation (36)) is obtained, which not only retains the advantages of the simple pretreatment process and high calculation efficiency of Algorithm 2, but also improves the calculation accuracy relatively. Through a large number of numerical calculation and engineering measured data, the rationality of the stress calculation value correction method proposed in this paper is verified, which can provide some references for similar projects.

## 8. Conclusions

In this paper, the equivalent inclusion theory is introduced into the elastic constant calculation of the equivalent pile foundation to achieve the anisotropic effect of the equivalent pile foundation, and it is realized by programming. By comparing the three algorithms with the measured values of the project, the following conclusions are obtained.

(1)The calculation results of the anisotropic pile foundation algorithm (Algorithm 3) based on the equivalent inclusion theory are closest to the measured values, and the relative error can be reduced by 10%~40% compared with the isotropic equivalent algorithm (Algorithm 2);(2)Algorithm 2 has the least difficulty in the pretreatment process, the highest efficiency, but the lowest accuracy. If Algorithm 2 is adopted for pile foundation, the first principal stress in the strong confined zones of concrete on pile foundation shall be multiplied by a coefficient. If the selected feature point is more than or equal to 0.5 m away from the free surface, the recommended value of the correction coefficient is $\alpha_{1}$, otherwise it is $\alpha_{2}$, and the variation range is from 0.72 to 0.76 and 1.32 to 1.36, respectively, and the calculation efficiency is actually improved.

## Figures and Tables

**Figure 1 materials-13-03815-f001:**
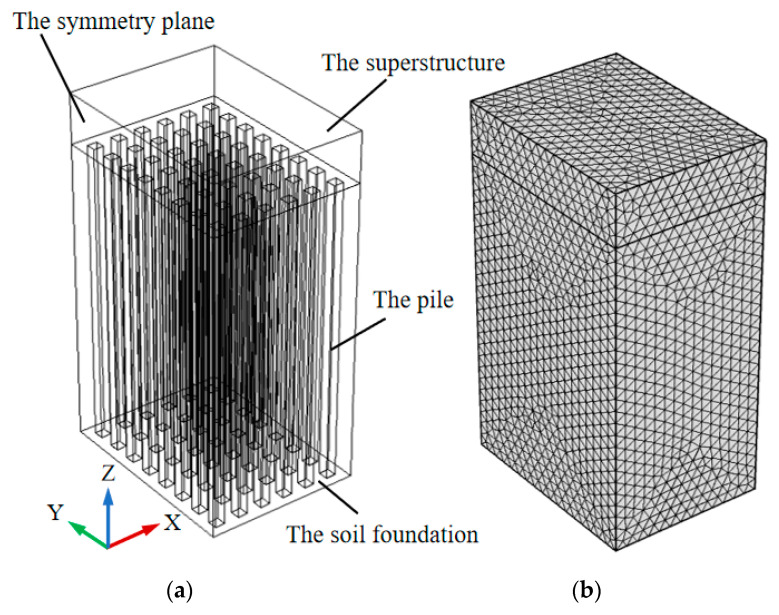
Calculation model and calculation grid diagram. (**a**) The whole model; (**b**) elements of the model.

**Figure 2 materials-13-03815-f002:**
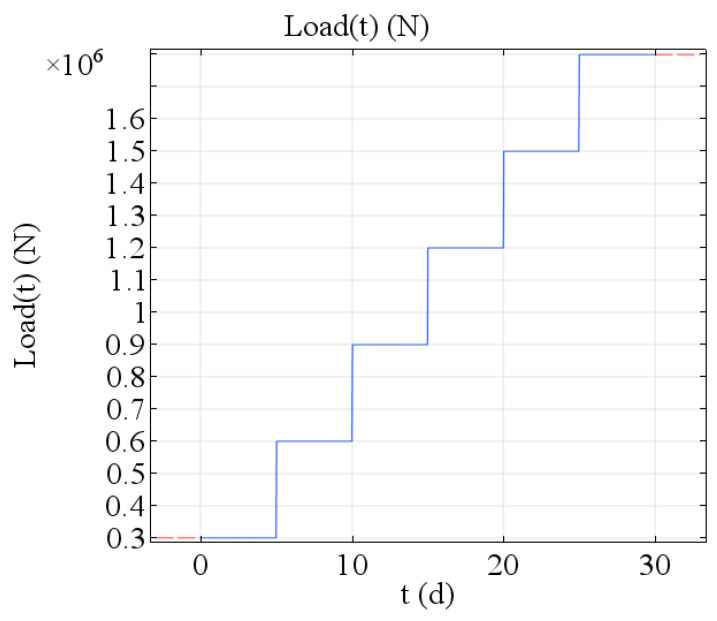
Schematic diagram of a step-by-step load.

**Figure 3 materials-13-03815-f003:**
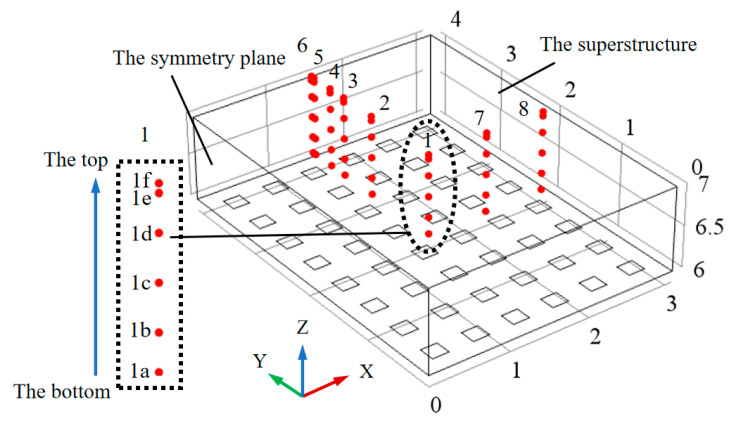
Schematic diagram of feature points selection location.

**Figure 4 materials-13-03815-f004:**
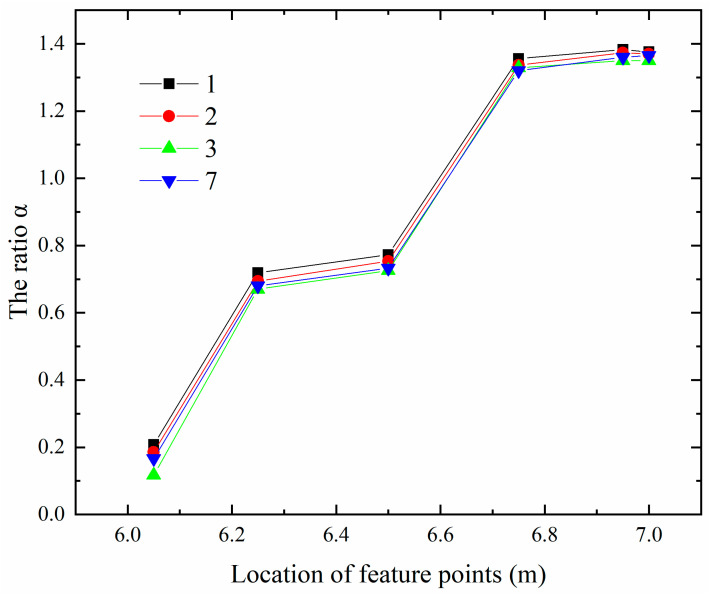
Ratio diagram of the maximum values of the first principal stress at the feature points in series 1, 2, 3, 7 in the vertical direction.

**Figure 5 materials-13-03815-f005:**
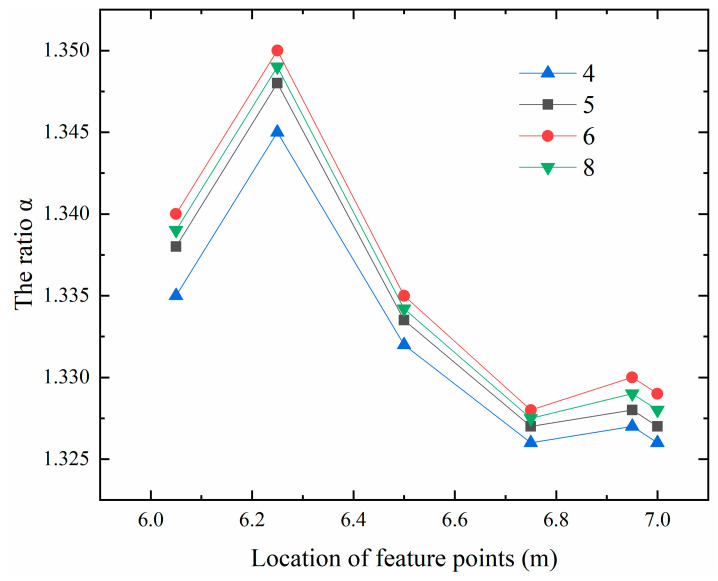
Ratio diagram of the maximum values of the first principal stress at the feature points in series 4, 5, 6, 8 in the vertical direction.

**Figure 6 materials-13-03815-f006:**
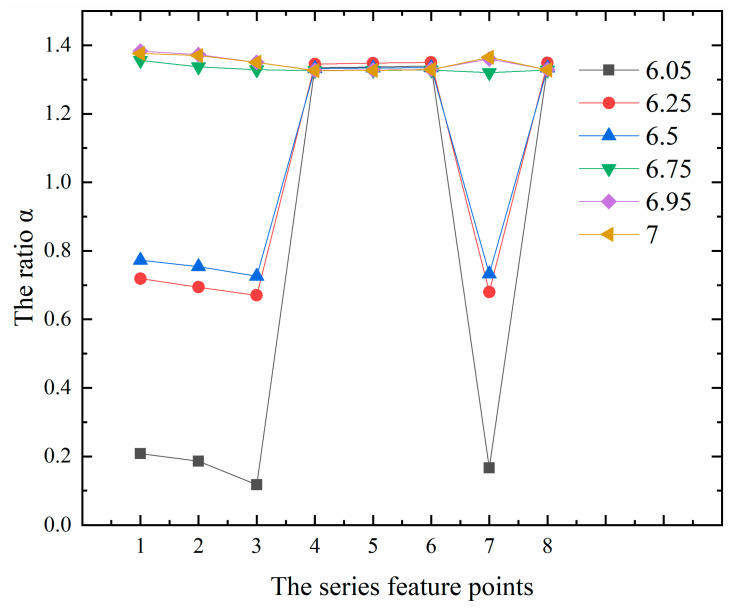
Ratio diagram of the maximum values of the first principal stress at the feature points in series 1, 2, 3, 4, 5, 6, 7, 8 in the horizontal direction.

**Figure 7 materials-13-03815-f007:**
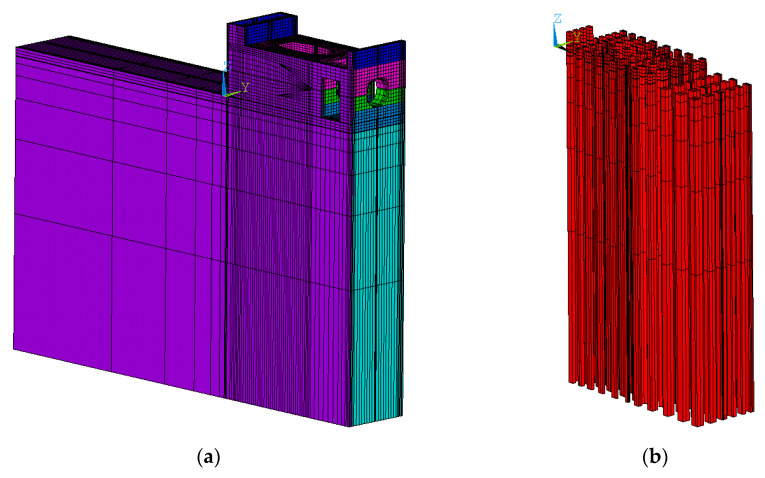
Finite element model and feature point location of the inlet section of Xiepu pump station. (**a**) The whole model; (**b**) elements of the pile; (**c**) elements of inlet section; (**d**) feature point 1.

**Figure 8 materials-13-03815-f008:**
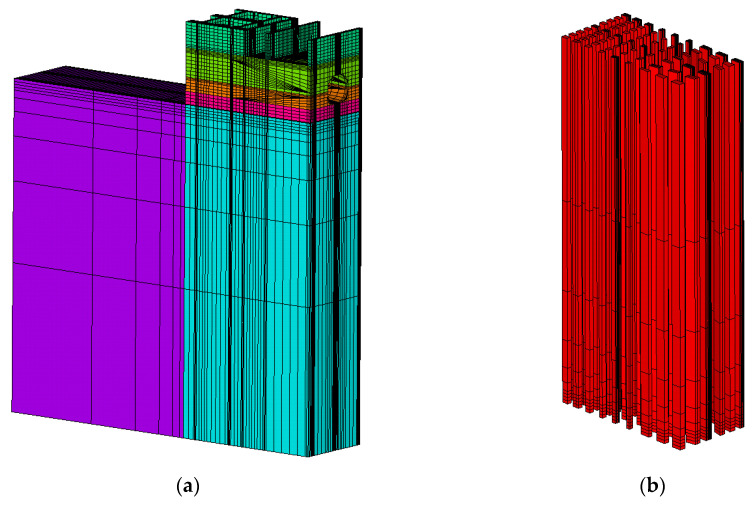
Finite element model and feature point location of the outlet of Xiepu pump station. (**a**) The whole model; (**b**) elements of the pile; (**c**) elements of outlet section; (**d**) feature point 2.

**Figure 9 materials-13-03815-f009:**
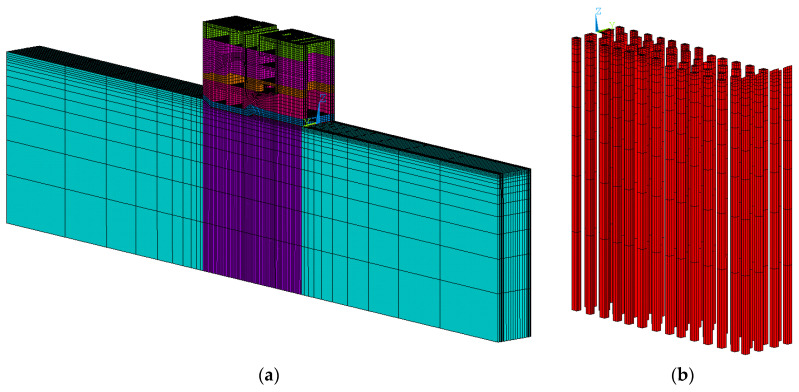
Finite element model and feature point location of the outlet section of Lianghu pump station. (**a**) The whole model; (**b**) elements of the pile; (**c**) elements of drainage structure; (**d**) feature point 3; (**e**) feature point 4.

**Figure 10 materials-13-03815-f010:**
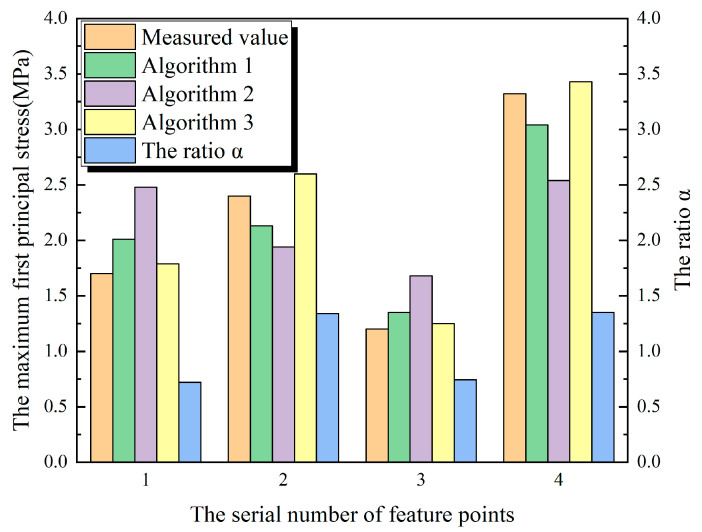
Comparison of three algorithms with engineering measured values.

**Figure 11 materials-13-03815-f011:**
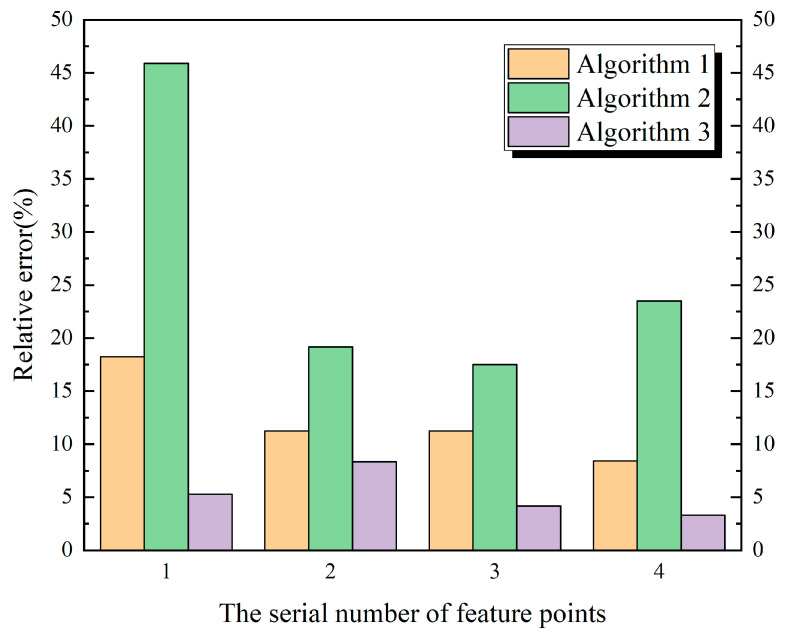
Relative error of three algorithms based on measured values.

**Table 1 materials-13-03815-t001:** The main mechanical parameters of concrete and soil.

Category	Elasticity Modulus *E_0_**/* MPa	Density ρ/ kg/m^3^	Poisson’s Ratio *μ*	Linear Expansion Coefficient *α*/ 10^−6^/K
Concrete Structure	28,000.00	2261.00	0.167	9.48
Pile	28,000.00	2261.00	0.167	9.48
Silty Clay	10.00	1830.00	0.30	8.00
Equivalent Pile Foundation (*n* = 0.156)	4209.00	1895.00	0.280	8.22

**Table 2 materials-13-03815-t002:** Maximum values of the first principal stress of the series feature points under three algorithms (MPa).

Point Value Series	1	2	3	4	5	6	7	8
a	0.00278	0.00095	1.30000	2.20000	−0.00044	0.01837	0.01300	0.42500
	0.00894	0.01210	1.01000	1.77612	0.03599	0.01533	0.04132	0.33607
	0.00186	0.00225	1.35000	2.38000	0.00421	0.02047	0.00690	0.45000
b	0.03939	0.04154	0.49930	0.54175	0.08690	0.18244	0.10204	0.31978
	0.04876	0.06204	0.40472	0.43066	0.11949	0.14536	0.14888	0.26640
	0.03506	0.04306	0.54556	0.58139	0.08006	0.19551	0.10124	0.35938
c	0.09100	0.10100	0.23800	0.24000	0.14767	0.19845	0.18000	0.24200
	0.10996	0.13528	0.18748	0.20075	0.19936	0.15648	0.23602	0.19637
	0.08500	0.10200	0.25000	0.26800	0.14473	0.20844	0.17300	0.26200
d	0.15179	0.14367	0.17196	0.15953	0.16462	0.17532	0.20205	0.17408
	0.10275	0.11155	0.13235	0.12187	0.11908	0.13606	0.17027	0.16357
	0.13932	0.14914	0.17563	0.16184	0.15825	0.18041	0.22476	0.21714
e	0.33000	0.25800	0.20000	0.20200	0.21326	0.19899	0.17200	0.11000
	0.22849	0.18937	0.16566	0.16015	0.16246	0.16377	0.13309	0.08653
	0.31600	0.26000	0.22000	0.21300	0.21932	0.21732	0.18100	0.11500
f	0.40500	0.29300	0.22000	0.22200	0.24437	0.21433	0.18300	0.08000
	0.25000	0.22628	0.14846	0.14823	0.17797	0.17157	0.15385	0.06401
	0.34400	0.31000	0.19700	0.19700	0.24017	0.22751	0.21000	0.08500

Note: The three data from top to bottom obtained from the intersection of each row and each column are the calculation results of Algorithm 1, Algorithm 2 and Algorithm 3, respectively.

**Table 3 materials-13-03815-t003:** Maximum values of the first principal stress at feature points of pouring blocks in different projects.

Engineering Project	Xiepu Pump Station		Lianghu Pump Station	
Feature Points	1	2	3	4
Distance from Free Face (m)	1.30	0.20	1.0	0.11
Distance from the Contact Surface of the Foundation	2.69	1.60	4.70	1.75
Measured Values	1.70	2.40	1.20	3.32
Calculated Value of Algorithm 1 (MPa)	2.01	2.13	1.35	3.04
Calculated Value of Algorithm 2 (MPa)	2.48	1.94	1.68	2.54
Calculated Value of Algorithm 3 (MPa)	1.79	2.60	1.25	3.43
The Ratio α	0.72	1.34	0.74	1.35

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
