# Peer review of "Study on the Calculation Method of Stress in Strong Constraint Zones of the Concrete Structure on the Pile Foundation Based on Eshelby Equivalent Inclusion Theory"

_materials, 2020, doi:10.3390/ma13173815_

Round 1

Reviewer 1 Report

I think that the paper is not ready for publication.

It contains several portion of the text that resembles more to an ordinary text book description rather than a scientific research article. I think that the research described is not well organized : it contains more than 25 figures, which make it more suitable for a book chapter.

The state of the art is not properly described and as a consequence the novelty of the work does not emerge at all with respect to what has been done in the literature so far.

I think the research does not meet the scientific standards of a journal such as Materials. As a consequence I cannot recommend its publication.

Reviewer 2 Report

Dear Autors,
I enjoyed reading this paper and found the results interesting. The introduction is clear and frames the need for this research well. The literature review is clear, although it could be more extensive. The methods used in the paper are analitical and numerical methods. The methods are appropriate for the work and clearly described. The analysis of the results is sufficient. The authors report the outcomes and provide analysis of the results. The article has the correct structure. English language and style are fine. Reference to figures and tables has been done correctly. The figures and tables are clear. I have only two comments on the manuscript:

1) Please provide more information on the numerical model (type of finite element, applied shape functions, what program was used for the calculation).
2) Line 326 move to the next page.

To conclude: the article is suitable for publication in its current form after minor corrections.

Reviewer 3 Report

The manuscript is not acceptable in its present form and it need major revision by careful addressing of the following points (attachment) and English language correction.

Reviewer 4 Report

The paper by Yuan et al. is in principle interesting, well documented and well written. It can deserve in principle publication on Materials, after the following minor revisions:

1) Lines 36-43. This period is too long. Perhaps, it’s better to separate it in two or more sentences.

2) Lines 43-46. This sentence is not clear because of English. Please verify.

3) Lines 48-50. This sentence is not clear because of English. Please verify.

4) Figures 4-11. Too many figures of the same type that are useless and makes the reading difficult. The authors have to reduce the number of figures, summarizing the results in a table or reporting them in the main text.

5) Figures 12-19. The same comment applies also here.

6) Lines 427-429. This sentence is not clear because of English. Please verify.

7) Lines 435-437. This sentence is not clear because of English. Please verify.

8) Lines 481-482. A reference should be added.

9) Lines 493-514. Place the figures after the text describing them.

10) Lines 532-533. Place Table 2 after the text describing it.

Round 2

Reviewer 1 Report

I appreciate the efforts of the authors, but I still think that the paper does not meet the criteria of innovation, impact and scientific soundness for a journal such as Materials. 

Reviewer 3 Report

The article has been revised very well and I have no further comments. I recommend the article for acceptance.